# Overview of the Incorporation of Legumes into New Food Options: An Approach on Versatility, Nutritional, Technological, and Sensory Quality

**DOI:** 10.3390/foods12132586

**Published:** 2023-07-03

**Authors:** Helena de Oliveira Schmidt, Viviani Ruffo de Oliveira

**Affiliations:** 1Institute of Food Science and Technology, Federal University of Rio Grande do Sul (UFRGS), Porto Alegre 91501-970, RS, Brazil; helena_schmidt@hotmail.com; 2Postgraduate Program in Food, Nutrition and Health, Nutrition Department, Federal University of Rio Grande do Sul (UFRGS), Porto Alegre 90035-003, RS, Brazil

**Keywords:** pulses, protein functionality, technological processing, flours, legume products

## Abstract

Consumers are more aware and demanding of healthy food options, besides being concerned with environment-friendly consumption. This paper aims to evaluate nutritional, technological, and sensory characteristics of legumes and their products’ quality and versatility, considering potential applications in new food options. Legumes are foods that have a recognized nutritional group since they have high protein and fiber content. However, their consumption is still somehow limited for some reasons: in some countries it is not easy to find all the species or cultivars, they need an organization and planning before preparation since they need soaking, and there is the presence of antinutritional factors. Due to the different functionalities of legume proteins, they can be applied to a variety of foods and for different purposes, as grains themselves, aquafaba, extracts, flours, brans, and textured proteins and sprouts. These products have been inserted as ingredients in infant food formulations, gluten-free foods, vegetarian diets, and in hybrid products to reduce food costs as well. Foods such as bread, cakes, cookies, meat analogues, and other baked or cooked products have been elaborated with nutritional, technological and sensory quality. Further development of formulations focused on improving the quality of legume-based products is necessary because of their potential and protein quality.

## 1. Introduction

There is a wide variety of legumes: beans, soybeans, peas, lentils, chickpeas, lupins, carobs, and peanuts, which can be purchased green, dehydrated, in pods, sprouted, or in canned form [1].

Pea (*Pisum sativum* L.) originating in the Mediterranean, especially in the Middle East, is one of the oldest domesticated crops in the world [2]. According to Faostat data, around 32 million tons of dried and green peas were produced worldwide in 2021. Soybean (*Glycine max*) has Asian origins and is one of the most versatile and nutritionally important legumes in the world [3]. Bitocchi et al. [4] cited that the wide geographic distribution of bean species reflects the diverse patterns of adaptation to the different ecological niches in which the various species evolved, which favors planting in the most different environments. Chickpea (*Cicer arietinum* L.) is another legume with an important role in world food and is one of the most cultivated [5]. Lentil (*Lens culinaris*) is also a legume and it is estimated that in the year 2020 more than 6 million tons of lentils were produced in the world. Among the world’s largest producers, Canada and India can be mentioned, and much of this production is exported [6].

This food group is considered important to ensure the quality of the diet, since they are important sources of nutrients such as proteins, lipids, carbohydrates, vitamins, minerals, and fibers, in addition to phytochemicals [7,8].

Legumes have nutritional, sensory, technological, and functional qualities such as viscosity, water and oil absorption capacity, foam formation, and emulsification. These functional properties have drawn attention as well, since they can favor the use of legumes as ingredients in many food options. This can be through the legume itself or its products, such as extracts, aquafaba, flours, brans, sprouts, and textured proteins.

Legumes could provide greater care for the environment and sustainability since they are a less environmentally damaging option when compared to those from animal origin. They are also less expensive than animal proteins and have nutritional quality as well. However, the search for quality foods and protein digestibility remains a concern and makes the diversification of protein sources in the diet a necessity [9,10].

Some papers have already mentioned the use of legumes, studied cereal-based foods enriched with legume flours, or studied extraction, functional properties, and effects of polyphenols on structural properties of legume-derived protein. Studies about physicochemical and functional properties of legume ingredients and their functional role in food product development have been published recently as well [11,12,13,14]. However, this current study brings an approach on the technological, versatility, and sensory aspects of new food options. Therefore, this paper evaluates nutritional, technological, and sensory characteristics of legumes and their products’ quality and versatility, considering potential applications in new food options.

To carry out this study, a narrative review was conducted. The selection of the papers was performed by two researchers who followed an initial set of studies built through searches in several important databases: Scopus, Science Direct, Web of Science, Springer Link, Gale, Technology Research Database, Cochrane, CAB Direct, PubMed, Lilacs (Latin American and Caribbean Literature on Health Sciences), and Capes Portal, between February of 2020 and December of 2022.

The identification of the studies was performed, and the papers were selected considering the following keywords: “legumes”; “pulses”; “legumes flours”; “pulses flours”; “legumes proteins”; “protein functionality”; “technological properties”; “nutritional and technological properties”; “texture-modified foods”; “sensory quality”; “sensory properties”; “legumes plant-based”; “legumes products”; “new food options”.

All the titles and abstracts of the articles were read in order to confirm if they addressed the study theme, and those that met all the inclusion criteria were read in their entirety. The targeted data included authors, legumes, food made, and outcomes. The inclusion criteria were (1) being compatible with the main subject; (2) free, online, digital access available for full reading and download. The exclusion criteria were (1) not being compatible with the main subject; (2) not addressing the topic of interest; (3) being an animal study; (4) being a dissertation, book, thesis, or literature review.

## 2. Nutritional Quality of Legumes

Legumes consist mainly of carbohydrates (15–68%), proteins (15–40%), and dietary fiber (15–35%) and, depending on the legume, a varied content of lipids, especially soybean and chickpea. Cellulose is considered an important fiber component, found mainly in beans, peas, lentils, and green beans [15], comprising 30–43% of the total fiber. Resistant starch is also present in peas and beans mainly; this fraction may comprise 30.3% and 24.2% of the total fiber, respectively [16]. These legumes are also considered good sources of vitamins like B-complex vitamins (thiamine, riboflavin, and folates), minerals (iron, zinc, copper, and manganese), and antioxidant compounds like tannins, phenolic acids, flavonoids, and isoflavones, in addition to carotenoids [16,17,18,19].

In legumes, the high amounts of protein deserve to be highlighted, which can vary from 20% in peas to 38–40% in soybeans and lupines [20,21]. Legume proteins are classified into albumins, globulins, glutelins, and prolamins according to their solubility, and the main proteins found in most legumes are globulins and albumins, including enzymatic proteins, protease inhibitors, amylase inhibitors, and lectins [20,22]. Despite having high protein content, proteins from legumes are still mentioned as inferior to those from animal [23,24], since the protein quality of food is evaluated based on the digestibility of proteins and amino acids, in addition to the amount of essential amino acids in relation to nutritional needs [8].

The digestibility of legumes is lower than that of cereal proteins and this inferiority can be attributed to several factors that are related to the protein’s structure and functionality, such as relative solubility, compartmentalization, bean proteins structure compactness, cell walls permeability, and seed protective coating, in addition to the presence of substances in the seed coat, such as tannins and phytic acid, which can form insoluble complexes with amino acids, making them unavailable for absorption [25,26]. This fact is especially important since, in some countries, legumes are used in association with cereals to minimize this inadequacy, in which legume proteins are complemented with cereal proteins, which are low in lysine but contain adequate amounts of methionine and cysteine [26]. This union is a promising source of dietary protein, especially for a population without many financial resources [10,27].

Despite having nutritional, sensory, and versatility advantages in cooking (Figure 1), legumes have compounds that can negatively affect their nutritional value and hinder the protein’s digestibility when consumed. For Hall et al. [16], phytates, trypsin inhibitors, lectins, and polyphenols, in addition to flatulence factors such as oligosaccharides, are antinutrients already known in legumes. These antinutrients can be reduced by discarding hydration water and minimized by thermal processing [28]. Shi et al. [29] demonstrated that lectins, phytates, and oxalates, also present in legumes, were substantially reduced by soaking and cooking. The elimination degree of antinutritional factors will depend on how soaking and temperature are carried out, in addition to water pH, the type of legume, and the solubility properties of such components [28].

Lentil-based foods have been successfully produced and marketed in recent years. Lentils are known to induce severe allergic reactions; however, it is currently unknown whether new lentil-based pasta retains the same allergenic potential as lentil seeds. Valdelvira et al. [30] studied the allergenic content of lentil food mass compared to lentil seeds by immunoassay. The effect of boiling processing was also analyzed. The results showed that the food lentil mass has a significant allergen content close to the general allergen content observed for lentil seeds. Both lentil pasta and lentil seeds were similarly affected by boiling, transferring allergens to the cooking water. Valdelvira et al. [31] also analyzed the allergenic content of chickpea mass. Chickpea pasta showed an important content of IgE binding proteins and chickpea allergens: 7S globulin, 2S albumin, LTP, and PR-10, similar to hydrated chickpea seeds and cooked. During boiling, more allergens from chickpea pasta were transferred to boiling water than chickpea seeds.

With regard to bioactive compounds, legumes have several types of compounds depending on the variety of species, color, and type of processing of the legume. For example, lupine seeds contain low amounts of carotenoids and high concentrations of tocopherols and phenolics. However, the chemical composition of Andean lupine flour was modified by the technological processes (debittering, extrusion, and spray-drying) applied [32,33]. After debittering, the tocopherol content increased slightly, while the carotenoid content remained practically unchanged and the phenolic concentration decreased markedly. Extrusion did not modify tocopherols, it marginally increased phenolics, but slightly reduced carotenoid concentration. Spray drying dramatically decreased tocopherols, carotenoids, and phenolics. The high content of free antioxidant compounds in these lupine flours suggests their possible high availability during digestion [32].

## 3. Legumes’ Germination: Antinutritional Factors and New Products Options

Germination, also known as sprouting, is the process of soaking the legumes in water and keeping them in moist conditions until they begin to germinate [34]. Legumes’ germination has become popular as a healthy diet proposal due to the positive effects of the nutritional and sensory properties of such food [35].

B-complex vitamins, for example, can increase in germinated grains, such as lentil sprouts, of which some studies already found a significantly higher amount than in dry lentil seeds [36]. The germination conditions were air temperatures of 25 ± 0.5 and 20 ± 0.5 °C day/night; the relative humidity was maintained at 60% (day) and 80% (night). Ninety-six hours of germination process had the best results for folate in legumes. The bioavailability of minerals can also increase in germinated legumes, as well as cause an increase in phytochemicals and bioactive peptide levels. Furthermore, germination can improve the digestibility and bioaccessibility of legume seed proteins.

Chickpea germination for 72 h followed by boiling, drying, and peeling was considered a useful technique for cooking or as a thickener for follow-on infant formulas fortified with minerals and vitamins [37]. Ohanenye et al. [38] also reported lower amounts of phytates, tannins, and trypsin inhibitors in different germinated legumes. In addition, germinated chickpeas and green peas had a reduction in flatulence-related oligosaccharides, as well as phytic acid and tannins [34].

Bresciani and Marti [39] evaluated the incorporation of germinated legumes in breads for nutritional enhancement and observed that the partial replacement of wheat flour by 15% of sprouted chickpea flour demonstrated higher protein content in the breads. This can suggest that germination is an economical and effective way to improve the nutritional value of legumes [40,41]. It is worth pointing out that the germination process was in dark conditions at 24 ± 1 °C and 80% of relative humidity for 96 h.

Perri et al. [42] germinated lentils for 24 h under 25 °C, which later developed into flour that could be inserted into fermented bread dough and showed structural and nutritional improvement, with good sensory attributes. Guardado-Félix et al. [40] observed that the addition of 15% of germinated chickpea flour to wheat flour allowed the production of yeast-leavened breads with appropriate functional and sensory properties and had more lysine and better protein quality when compared to the control.

Ahure and Ejoha [43] evaluated cookies made with soybean sprouted for 96 h and pointed out that the cookies were considered richer sources of proteins, ashes, and crude fat. These authors considered their cookies to be interesting options to increase protein content and minimize the challenges of protein deficiencies among children in the low-income group, as well as for gluten-allergic individuals. Legume sprouts seem to also be a promising strategy to improve the sensory quality of new foods, especially for those who follow vegetarian or vegan diets (Figure 2).

## 4. Use of Legume Flours and Other Products

Legume flours have been inserted in diets with or without gluten to make breads [44], cookies [45], cakes, muffins [46], pasta, and other products. Legume flour use can be partial or total. Many of these uses also aim to reduce the cost of preparation and improve acceptability and nutritional quality, thus favoring the complementation of the essential amino acid content with cereals, which has become particularly interesting to improve the protein quality of foods that were previously exclusively based on cereals [22].

The addition of legume flour also favors viscosity, dough volume, specific bread volume, cake tenderness, and texture of bakery [47]. Legume gelling has been applied to foods such as puddings, mousses, soups, creams, sauces, and gels [48]. The gel-forming properties of legumes have been studied for the management of dysphagia as well, in addition to the increase in nutritional value, since these patients often find themselves in a state of malnutrition. Chickpea is an example of a promising legume that has been used for such purpose [49]. Chickpea, lentil, soy, pea, and bean flours have been the most used in this kind of food. One of the limitations is that not all studies detailed the particle size of these flours used, which may lead to low miscibility of all foodstuffs used in the preparation and may affect sensory quality of the final product [50].

Giuberti et al. [51] made spaghetti with rice flour and bean flour. The use of legumes as functional ingredients in gluten-free bakery and pasta products have been extensively investigated [43]. Angioloni and Collar [52] mentioned that the use of legume flour is technologically a challenge, since more than 15% of legume flour can make it difficult to dilute gluten, which usually requires the addition of structural agents. In addition, the association between legumes and cereal proteins should favor the formation of a viscoelastic mass, the incorporation of air, and gas retention during fermentation to avoid a weak crumb structure.

Hoehnel et al. [53] evaluate the nutritional profile of a high-protein hybrid pasta formulation. In the formulation, a combination of three high-protein ingredients from buckwheat, faba bean, and lupin was used to partially substitute wheat semolina. The nutritional value of the pasta was assessed in comparison to regular wheat pasta as a reference, with a focus on protein quality. All measures indicative of protein quality determined in this study (amino acid composition, in vitro protein digestibility, in vivo nitrogen balance, protein efficiency ratio) conclusively suggested improved protein quality of high-protein hybrid pasta compared to reference wheat pasta. The results also indicate that, specifically, the combination of pseudo cereal and legume high-protein ingredients to replace wheat semolina is beneficial in achieving a balanced amino acid profile. In addition to its enhanced nutritional profile, the high-protein hybrid pasta has been shown to possess technological and sensory quality similar to reference wheat pasta, which identifies high-protein hybrid pasta as an attractive alternative to regular wheat pasta in currently consumed diets. Furthermore, high-protein hybrid pasta and formulations of its kind represent an increased potential of wheat-based staple foods to contribute to a sufficient intake of high-quality protein in future predominantly plant-based diets [53]. In the study by Betrouche et al. [54], fava bean flour presented a protein content of about four times that of rice flour.

Komeroski et al. [55] evaluated breads made with chickpea flour, cassava flour associated with whey protein, and found the highest ash content in the breads with chickpea, which occurred probably due to the higher amount of chickpea flour. The same authors highlighted some details about color, since this attribute received higher scores in their sensory evaluation. Although wheat bread is considered the gold standard for quality, in their study the wheat flour treatment was not the best in all sensory attributes, which means that the yellowish color that chickpeas flour brought to the breads, quite different from the color of the wheat bread treatment, may have pleased the assessors. Guimaraes et al. [56] observed that soybeans can cause a desirable browning in the crust and crumb of the breads as well.

Pasqualone et al. [57] studied lentil flours and observed that in the extrusion process the samples showed reduced tendency to retrograde and were suitable for the formulation of bakery products, making gluten-free products less prone to hardening. In addition to celiacs, these new products based on proteins from legumes could satisfy the growing market for vegan consumers, which also have been used as an ingredient substitute for whey protein [51].

Joehnke et al. [58] studied the nutritional and antinutritional properties of lentil protein isolates compared to whole-seed flours. They found that the lentil protein isolates had increased protein content and reduced levels of trypsin inhibitor activity (around 81–87%) when compared to whole-seed flours. Furthermore, digestibility was also improved. For pepsin, the digestibility of the isolates was increased by 35–49%.

Du et al. [59] studied flours from different legumes and observed that for each functionality parameter, such as water absorption index, oil absorption capacity, and emulsion activity, a particular type of legume flour distinguished itself. For example, lentil flour was denser than other legume flours, chickpea flour had the highest water absorption index and black eye bean flour had the lowest value, black bean flour showed higher oil absorption capacity, small red bean flour had the highest emulsion activity and chickpea flour the lowest activity, and chickpea flour had low pasting temperature and degradation.

Evangelho et al. [60] studied the protein hydrolysates of black bean treated with alcalase and pepsin. Black bean protein hydrolysates treated with alcalase showed greater hydrophobicity and better emulsion stability than those obtained by pepsin digestion, which is important for the food industry in relation to liquid food applications where phase stability is desirable.

Ma et al. [61] evaluated salad dressing and lentil, chickpea, and pea protein isolate. Physical properties and rheological behavior of salad dressings made with lentils, peas, and chickpeas to replace egg yolks in different oil concentrations were investigated. Proteins prepared from legumes can be promising value-added substitutes for emulsion-type food products that could be used by the food industry to develop salad dressings focusing on low-cholesterol and hypoallergenic diets.

## 5. Using Other Products from Legumes

### 5.1. Aquafaba

Recent studies have shown that the viscous liquids that can be drained from cooked or canned legumes produce stable foams, emulsions, and gels. This liquid, called aquafaba, can be used as a gluten-free and cholesterol-free vegan rheological additive in many food products such as egg-free mayonnaise, meringue, mousse, whipped cream, ice cream, emulsified sauces, cocktails, and confectionery products. A recent study with chickpeas, beans, lentils, soybeans, and peas investigated the use of such ingredients [62,63].

Chickpea in aquafaba was studied by Mustafa et al. [64]. Aquafaba was added as a foaming and texturing agent to substitute egg white in sponge cake. Color, appearance, and texture of aquafaba sponge cake (without egg) were similar to sponge cake made with egg white. The aquafaba cake exhibited less elastic and less cohesive properties than the egg white cakes.

Stantiall et al. [63] also evaluated aquafaba; however, the authors used haricot, beans, chickpeas, lentils, and split peas in aquafaba. Lentils had greater foaming capacity, while chickpeas had the greatest gelling capacity. Sensory analysis of aquafaba meringues showed low acceptance for the flavor of meringues made with beans and lentils. High acceptance was found for chickpeas and peas, similar to egg white.

Shim et al. [65] evaluated ten samples of chickpea aquafaba and observed that the protein content ranged from 22.7% to 26.8%, and among amino acids, alanine was the main one found. Most aquafaba proteins are of low molecular weight, being identified as heat-soluble hydrophilic species and heat shock proteins with well-known heat stability [59]. Aquafaba has several nutrients, in addition to proteins, such as simple sugars, polysaccharides, minerals, saponins, and phenolic compounds, which contribute to foaming capacity, emulsibility, gelling, and thickening properties.

Aquafaba’s physical properties, such as pH, density, viscosity, water, and oil absorption capacity, determine the functional properties of this new product. The lower pH of aquafaba can lead to an increase in foaming and emulsification properties [62,63,66].

Many consumers are concerned about aquafaba’s oligosaccharides, as such compounds can cause unpleasant gastrointestinal symptoms such as flatulence, bloating, diarrhea, and abdominal pain [67]. However, the degree of reduction of antinutrients in aquafaba depends on its structure, seed/water ratio, thermal processing practice, time, and temperature. Therefore, prior to preparing aquafaba, soaking dried legume grains in water can reduce the concentration of antinutritional components, including α-galactosides, saponins, minerals, phytic acid, oxalate, and proteolytic enzyme inhibitors, which are partially or fully solubilized in the solution of immersion [68]. Therefore, for the preparation of aquafaba, the practice of cooking the beans in water under high temperature and pressure is encouraged to minimize or inactivate heat-sensitive antinutritional compounds, such as trypsin inhibitors, and to decrease the content of phytic acid and α-galactoside. For He et al. [69], the concentration and activity of such antinutritional factors in aquafaba are significantly lower than in raw legume seed.

Due to its functionality, aquafaba can be used in food products, such as vegan mayonnaise and butter [69], and gluten-free bakery products (biscuits, sponge cake, and bread), since it is a texture enhancer due to its gelling properties and water retention capacity [62].

### 5.2. Legumes as Substitutes for Animal Source Foods

The perspectives for consumption and marketing of legumes as substitutes for animal source foods also seem very promising due to the quality of the proteins in such grains, and the motivations are quite varied: for vegetarians, for nonvegetarians who respect animals, people who want to consume foods without cholesterol, or for religious or ethical reasons.

Ma et al. [61] evaluated that legume protein could be a helpful substitute for those who need to avoid eggs or emulsion food products, since some people are allergic or need to control their cholesterol. The authors demonstrated the feasibility of making salad dressings with similar physical properties to commercial dressings by varying levels of lentil, chickpea, and pea protein isolates.

Consumers can have more than one motivation to replace meat with legumes and as an ingredient in processed foods, because of their promising processing qualities, since legumes can be included in many convenient products. In addition, the increased consumption of processed legumes products could reduce the use of meat in many dishes [70].

Meat substitutes deserve to be further evaluated, especially regarding technological quality and acceptance, as some people who are interested in their consumption require that they be practical and have sensory characteristics close to the usual meat products such as appearance, texture, and general acceptability [71]. Kim et al. [72] evaluated hamburgers made with meat analogues, made with different legumes, and compared to soy (control). The cooked hamburgers containing legume proteins turned redder and had a higher cooking yield. Hamburgers with faba beans required less cooking time, while pea and lentil proteins used in vegetable hamburgers did not influence consumer preference for cooked appearance and overall flavor. Hamburgers made with faba beans had lower overall taste compared to control (soy). In contrast, the overall texture was lower for pea, lentil, and fava bean burgers compared to soy. Vegetable hamburgers with legume proteins are competitive with soy-based samples. The same authors observed that the texture seems to be an obstacle. Meat analogues have emerged with nutritional and sensory potential and as a suggestion for menu variety. Some consumers want preparations such as hamburgers, meatballs, croquettes, and fried products made with meat analogues [72,73,74]. Some options for meat analogues that have been studied include texturized vegetable protein, which is a dry product derived from soybeans, soy concentrate, mycoprotein, and modified defatted peanut flour [73].

### 5.3. Special-Purpose Legume Application

Legumes can also be used for special diets like for people with dysphagia (Figure 2). The main strategy to treat this problem is to use foods with modified texture and thick fluids, which allows the formation of a more cohesive food bolus that makes swallowing slower and safer [75]. The insertion of legume flours as a strategy for these people is promising, because they can make swallowing safer when they reach the proper consistency, thanks to the gelling properties of legume proteins. This gel-forming property of legumes has been widely studied in the management of dysphagia, in addition to the increase in nutritional value, such as the amino acid and fiber content of legumes, which is of utmost importance, since these patients are often in a state of malnutrition. Chickpeas are an example of a legume that has been used for this purpose [49].

Protein aggregates result in thickening, increased viscosity, and the formation of a gel, whereas interactions between the denatured protein and the gelatinized starch of the legume grain contribute to the increase in apparent viscosity [69]. Chickpea flour has already been used in doughs as well; since it is poorly shear-resistant, these characteristics are related to its high fat content. In addition, the lower tendency of retrogradation is an advantage in food products such as soups and sauces, which suffer loss of viscosity and precipitation as a result of retrogradation [59].

## 6. Technological and Functional Properties of Legume Proteins

Food made with legumes as ingredients can profit from the quality of the proteins of these grains, since they may have functional properties such as solubility, water and oil retention capacities, gelling, emulsification, and foaming, among others, which are known to improve food texture [76,77,78].

Several minimal nonthermal processing methods are being used to induce significant changes in protein conformation and modulation without altering its native characteristics. Pulsed light, high-pressure processing (HPP), irradiation, ultrasound, supercritical carbon dioxide, plasma technology, and pulsed electric field (PEF) are emerging technologies. HPP treatment induces denaturation and aggregation or gelling of proteins with high textural properties. The functional properties of legume proteins such as hydration, gelling, and emulsification make legume proteins also suitable for industrial applications [79].

Viscosity, water and oil retention capacity, and activity and stability of emulsions in legume preparations are influenced by the content of protein and soluble/insoluble carbohydrate [62,80], polysaccharide–protein complexes, lipids, coacervates, saponins, and phenolic compounds. Furthermore, different types of interactions among these molecules during the preparation of legumes and processing time determine the functional properties [62].

Water absorption during immersion followed by high-temperature treatment during cooking leads to hydration and protein denaturation and starch gelatinization, in addition to solubilization [64,81]. Changes in the coating of the grain depend on the pressure and temperature applied during soaking and cooking. Exposure to high pressures and temperatures and prolonged cooking times induces starch swelling and gelatinization [69]. Gels can be formed by heat application; however, overheating proteins at high temperatures (100 °C) can cause peptide bond scission that can impede gelling [82].

Soy protein has been widely used in the food industry for such purposes, in addition to peas, lentils, beans, and chickpeas [83,84]. In addition to providing functionality, legume proteins are also studied because their protein hydrolysates can exhibit health-promoting properties such as antioxidant activity. Evangelho et al. [60] studied black bean protein hydrolysates and observed that, when treated with alcalase, these hydrolysates can be used as emulsifying agents due to better emulsion stability.

According to their diverse functionalities and nutritional properties, legume proteins can be applied in a variety of foods and formulations and for different purposes. These functional properties make it possible to use legume proteins in the form of extracts, flours, textured proteins, and other derivatives in preparations such as soups, beverages, snacks, baked goods, or meat analogues [20]. Some improvements have been evaluated in some sensory tests of legume proteins to increase their use as promising ingredients [85,86].

The secondary structure of legume proteins is directly linked to the functional characteristics and changes that occur in protein conformation, for example, emulsifying and foaming properties of legume proteins. High-pressure processing exposes the larger sites, which significantly improves protein digestibility due to structural loosening and protein unfolding [79].

Different processing conditions can affect protein properties and functionality. In general, heat treatments such as cooking, microwave cooking, pressure cooking, and extrusion increase the digestibility of in vitro protein. In addition, an increase in cooking time was shown to be important for increasing in vitro protein digestibility (IVPD). Cooking resulted in better IVPD for lentils, chickpeas, peas, and soybeans, but soaking legumes before heating did not result in consistent effects. In microwave cooking, the lowest amount of energy (500 J/g) caused a significant increase in protein digestibility. A further increase in energy to 1000 J/g significantly improved protein digestibility. However, more energy input (1250, 1500, 1750 J/g) during microwave cooking did not significantly affect protein digestibility. Soaking before pressure cooking reduced processing time and positively affected bean protein digestibility, as IVPD improved. With regard to the baking process, apparently, the process of mixing, kneading, rising, and baking the dough reduced IVPD in relation to cooking, except for the red lentil. The extrusion of common bean, pea seed, broad bean, and kidney bean flour significantly increased the IVPD up to 87% [87].

## 7. Conclusions

Legumes have nutritional, technological, and sensory potential to be added to foods and become new options, and their low environmental impact should also be highlighted. However, some factors limit their use, in addition to the presence of antinutrients, which can be minimized through planning, hydration, and heat treatment. Germinated legumes have been used to improve the nutritional and sensory profile of preparations in addition to presenting lower levels of antinutritional factors.

Due to the different functionalities of legume proteins, they can be applied to a variety of foods and formulations and for different purposes, whether in the form of the grains themselves, aquafaba, extracts, flours, brans, and textured proteins. From a technological point of view, the incorporation of legumes in cereal-based products has been welcome, whether for celiac individuals or people sensitive to gluten, as a strategy that adds nutritional value and expands food options. However, dough rheology and the technological quality of breads, cakes, and biscuits still need to be further investigated.

Many people benefit from this increase in the consumption of legumes, such as people who do not consume gluten, vegans, vegetarians, allergic and dysphagic individuals, in addition to the world population that already has the habit of consuming such grains, which have culinary versatility. It is considered financially accessible and more sustainable than the consumption of animal source foods.

For future perspectives, our research group is focusing on evaluating the behavior of different legumes associated with other vegetable options in meat analogues, as well as chemical, technological, and sensory quality of these final products. 

## Figures and Tables

**Figure 1 foods-12-02586-f001:**
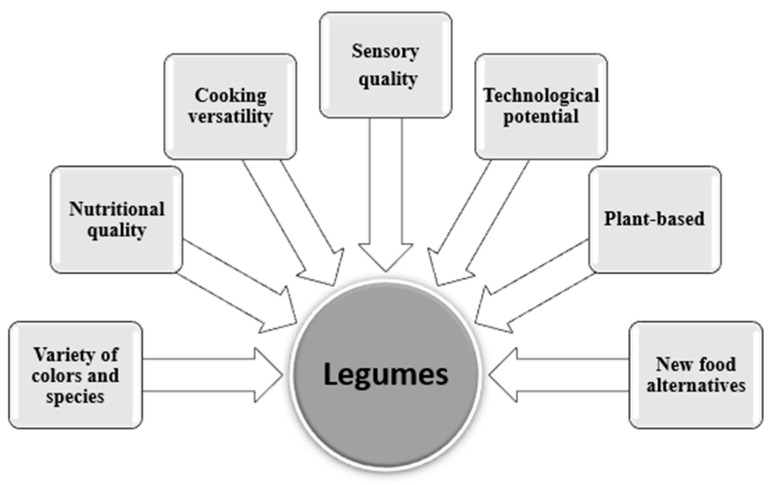
Positive aspects of legume consumption and new possibilities.

**Figure 2 foods-12-02586-f002:**
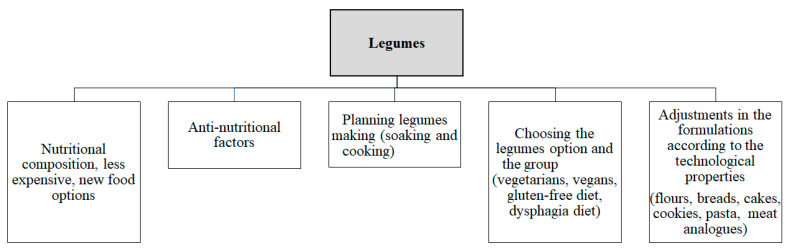
Planning and managing the use of legumes in new food options.

## Data Availability

Data is contained within the article.

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
