# Peer review of "Overview of the Incorporation of Legumes into New Food Options: An Approach on Versatility, Nutritional, Technological, and Sensory Quality"

_foods, 2023, doi:10.3390/foods12132586_

Round 1

Reviewer 1 Report

Manuscript ID: foods-2441646

Manuscript Title: Overview of the incorporation of legumes and their by-products into new food options: An approach on versatility, nutritional, technological and sensory quality

This paper made an overview of the incorporation of legumes and their by-products into new food options. The results and conclusions of this paper provide important theoretical basis and guiding significance for the comprehensive utilization of legumes. However, there are many problems with the content and format of this review paper; paper writing is not clear, summary ability is relatively weak.

 Comments:

1. The content of this paper is too broad and not focused enough. For example, it is enough to write about the progress on the legumes protein, or by-products.

2. In the introduction, some of the contents felt weak in keeping with the theme. For example, lines 55-77 are recommended to be deleted.

3. In the introduction, it is suggested that the author should add some related data of yield, consumption and varieties of legumes.

4. Corresponding to the contents in lines 72-82, it is suggested that the author should add some relevant table contents such as the nutritional content of different legumes varieties.

5. In Figure 1, it is suggested that the two aspects of positive aspects and anti-nutritional factors should be plotted separately.

6. The review and summary of relevant research progress in Part 3-5 is insufficient.

7. In the conclusion part, it is suggested that the author should add some words about the prospect of future research.

8. The format of some references is not standard, so proofread one by one.

Author Response

Dear Editor and Referees,

              We would like to thank the reviewers and the editor for your valuable time during the review process and the insightful comments which helped us to improve the quality of the paper. We have appreciated all the considerations about our manuscript and we agreed with all of them.                                                                                                                  We have added the suggestions and corrections pointed out by the referees. Questions and suggestions are in black and answers are in blue.  If something is not exactly as it was suggested, please let us know and we can write it again.

 Comments to Authors:Reviewer #1:

This paper made an overview of the incorporation of legumes and their by-products into new food options. The results and conclusions of this paper provide important theoretical basis and guiding significance for the comprehensive utilization of legumes. However, there are many problems with the content and format of this review paper; paper writing is not clear, summary ability is relatively weak.

Response: Thanks for your feedback, we have given our best on this paper, especially because our deadline was really short, but we really believe that this review will be useful for research fellows and students. All suggestions of our reviewer were accepted, we understand when other people read it, they may have a different perception and we believe that more people reviewing it and giving suggestions will improve the   paper quality.

The content of this paper is too broad and not focused enough. For example, it is enough to write about the progress on the legumes protein, or by-products.

 Response: Dear reviewer, we have tried to make it clearer in this 2º version, thanks for pointing this part could be improved.

In the introduction, some of the contents felt weak in keeping with the theme. For example, lines 55-77 are recommended to be deleted.

 Response: It was improved. Sorry! It was supposed to be the Material and methods to explain our papers searching. We have tried to make it clearer.

In the introduction, it is suggested that the author should add some related data of yield, consumption and varieties of legumes.

 Response: Thank you for the advice. We have added some data of yield, consumption, and varieties of legumes it in this new version as proposed by our referee.

Corresponding to the contents in lines 72-82, it is suggested that the author should add some relevant table contents such as the nutritional content of different legumes varieties.

 Response: It was added more nutritional content of different legumes variety, following our reviewer suggestion.

In Figure 1, it is suggested that the two aspects of positive aspects and anti-nutritional factors should be plotted separately.

 Response: It was corrected as our referee has suggested.

 The review and summary of relevant research progress in Part 3-5 is insufficient.

 Response: It was improved as our referee has suggested

In the conclusion part, it is suggested that the author should add some words about the prospect of future research.

 Response: It was added in the end of our discussion, close to conclusion.

The format of some references is not standard, so proofread one by one.

Response: It was corrected as our referee has suggested.

Reviewer 2 Report

The manuscript contains a literature review on the nutritional value and utilization of legume seeds in food technology. Several specific comments are provided below.

Lines 61-69: The main weakness concerns the selection of references, particularly important in a review publication. Articles that are significantly related to the topic have been omitted, e.g. such as the utilization of soy, pea, and lupin proteins in gluten-free bread.

Lines 72/73: It would be better to list the nutritional components in order of their highest content, such as "Legumes consist mainly of carbohydrates (15-68%), proteins (15-40%), and dietary fiber (15-35%)..."

Line 77: Please state specifically which vitamins and minerals.

Chapter 3 and others: It would be helpful to provide some details of the processes described, probably different authors used different parameters, for example, germination and other processes described; to what extent (by how many percent) there was a decrease in the content of antinutrients (and especially those causing unpleasant stomach sensations) when using the described processing methods?

Lines 161/162 and 172/173: These two sentences seem contradictory.

Line 181: Flour is not a by-product; it is intentionally produced, like e.g. wheat flour and other.

Line 309: Did you mean "pasta"?

English is readable and understandable.

Author Response

Dear Editor and Referees,

              We would like to thank the reviewers and the editor for your valuable time during the review process and the insightful comments which helped us to improve the quality of the paper. We have appreciated all the considerations about our manuscript and we agreed with all of them.                                                                                                                  We have added the suggestions and corrections pointed out by the referees. Questions and suggestions are in black and answers are in blue.  If something is not exactly as it was suggested, please let us know and we can write it again.

 Comments to Authors:

Reviewer #2:

Lines 61-69: The main weakness concerns the selection of references, particularly important in a review publication. Articles that are significantly related to the topic have been omitted, e.g. such as the utilization of soy, pea, and lupin proteins in gluten-free bread.

Response: We apologize to our reviewer for the expectation we created. We tried to improve this part, but since it was not a systematic review, we did not have the idea of including all the papers that address this topic. Our goal was to illustrate and bring the differences in products that can be made with legumes, ranging from hamburgers to breads, among other options. Besides, when we cross-referenced ‘protein’, ‘vegetable protein’ and ‘new food option’ in our search, papers on gluten appeared, which were inserted. We did not insert legumes specifically because the idea was not to favor any of them, but legumes in general. Anyway, we did a new search and included more papers, as our reviewer suggested.

Lines 72/73: It would be better to list the nutritional components in order of their highest content, such as "Legumes consist mainly of carbohydrates (15-68%), proteins (15-40%), and dietary fiber (15-35%)..."

Response: It was added, following our reviewer suggestion.

Line 77: Please state specifically which vitamins and minerals.

Response: It was improved as the referee has suggested.

Chapter 3 and others: It would be helpful to provide some details of the processes described, probably different authors used different parameters, for example, germination and other processes described; to what extent (by how many percent) there was a decrease in the content of antinutrients (and especially those causing unpleasant stomach sensations) when using the described processing methods?

Response: You are right, it was written in a poor way. We have improved and clarified it to avoid ambiguity.

Lines 161/162 and 172/173: These two sentences seem contradictory.

Response: It was corrected as our referee has suggested.

Line 181: Flour is not a by-product; it is intentionally produced, like e.g. wheat flour and other.

Response: Thank you so much for your attentive eyes, it really did sound much better. We corrected it in this new version.

Line 309: Did you mean "pasta"?

Response: It was improved as our referee has suggested.

Reviewer 3 Report

Dear Authors and Editors,

The article concerns a valuable and future-oriented topic. However, the purpose of the review should be clearly defined and it should be indicated what has been described so far in previous reviews on related topics, what is new in the work.

Introduction

What do you mean regarding “exclusion criteria were: (1) not available for complete reading”.  Please specify whether this means that the publication must be available in open access? or are other paid publications considered?

Before the purpose of the work, please specify what related reviews (e.g. Kaale et al. 2022; Binou et al. 2022; Wen et al. 2022; Keskin et al. 2022) have been published so far and mark what is new in the current work.

Technological and functional properties of legumes’ proteins

In the content of the work at the beginning of this chapter, it is worth giving what are the types and methods of isolation of legume proteins, and at the end of the chapter also how the functionality (Mulla et al. 2022) of these proteins can be increased, how the process conditions (Drulyte and Orlien, 2019) affect the properties of proteins.

Conclusion

At the end of the conclusion, it is worth stating what the authors anticipate are the future directions of research in the field of the discussed topic

Proposed references

Kaale, L. D., Siddiq, M., & Hooper, S. (2022). Lentil (Lens culinaris Medik) as nutrient-rich and versatile food legume: A review. Legume Science. https://doi.org/10.1002/leg3.169

Binou, P., Yanni, A. E., & Karathanos, V. T. (2022). Physical properties, sensory acceptance, postprandial glycemic response, and satiety of cereal based foods enriched with legume flours: a review. Critical Reviews in Food Science and Nutrition. Taylor and Francis Ltd. https://doi.org/10.1080/10408398.2020.1858020

Wen, C., Liu, G., Ren, J., Deng, Q., Xu, X., & Zhang, J. (2022, February 2). Current Progress in the Extraction, Functional Properties, Interaction with Polyphenols, and Application of Legume Protein. Journal of Agricultural and Food Chemistry. American Chemical Society. https://doi.org/10.1021/acs.jafc.1c07576

Keskin, S. O., Ali, T. M., Ahmed, J., Shaikh, M., Siddiq, M., & Uebersax, M. A. (2022, March 1). Physico-chemical and functional properties of legume protein, starch, and dietary fiber—A review. Legume Science. John Wiley and Sons Inc. https://doi.org/10.1002/leg3.117

Mulla, M. Z., Subramanian, P., & Dar, B. N. (2022). Functionalization of legume proteins using high pressure processing: Effect on technofunctional properties and digestibility of legume proteins. LWT158. https://doi.org/10.1016/j.lwt.2022.113106

Drulyte, D., & Orlien, V. (2019). The effect of processing on digestion of legume proteins. Foods8(6). https://doi.org/10.3390/foods8060224

Author Response

Dear Editor and Referees,

              We would like to thank the reviewers and the editor for your valuable time during the review process and the insightful comments which helped us to improve the quality of the paper. We have appreciated all the considerations about our manuscript and we agreed with all of them.                                                                                                                  We have added the suggestions and corrections pointed out by the referees. Questions and suggestions are in black and answers are in blue.  If something is not exactly as it was suggested, please let us know and we can write it again.

 Comments to Authors:

Reviewer #3:

The article concerns a valuable and future-oriented topic. However, the purpose of the review should be clearly defined and it should be indicated what has been described so far in previous reviews on related topics, what is new in the work.

Response: Thanks for your inspiring comments and advices, we have worked hard on this project/paper and we really believe that this review will be helpful for students and researchers. We also would like to thank our referee for sending the papers below, they were very helpful and it was very gentle from you.

Introduction

What do you mean regarding “exclusion criteria were: (1) not available for complete reading”.  Please specify whether this means that the publication must be available in open access? or are other paid publications considered?

Response: We completely agree with these suggestions and we have improved the paragraph, we hope you appreciate it in this new version.

Before the purpose of the work, please specify what related reviews (e.g. Kaale et al. 2022; Binou et al. 2022; Wen et al. 2022; Keskin et al. 2022) have been published so far and mark what is new in the current work.

Response: Dear reviewer, we completely agree and we have tried to improve it in this 2º version, thanks for pointing that this part could be improved.

Technological and functional properties of legumes’ proteins

In the content of the work at the beginning of this chapter, it is worth giving what are the types and methods of isolation of legume proteins, and at the end of the chapter also how the functionality (Mulla et al. 2022) of these proteins can be increased, how the process conditions (Drulyte and Orlien, 2019) affect the properties of proteins.

Response: Thank you for the advice. We have improved it in this new version as proposed by our referee.

Conclusion

At the end of the conclusion, it is worth stating what the authors anticipate are the future directions of research in the field of the discussed topic

Response:  We inserted at the end of discussion as “Future Perspectives. It was improved as our referee has suggested.

Proposed references

Kaale, L. D., Siddiq, M., & Hooper, S. (2022). Lentil (Lens culinaris Medik) as nutrient-rich and versatile food legume: A review. Legume Science. https://doi.org/10.1002/leg3.169

Binou, P., Yanni, A. E., & Karathanos, V. T. (2022). Physical properties, sensory acceptance, postprandial glycemic response, and satiety of cereal based foods enriched with legume flours: a review. Critical Reviews in Food Science and Nutrition. Taylor and Francis Ltd. https://doi.org/10.1080/10408398.2020.1858020

Wen, C., Liu, G., Ren, J., Deng, Q., Xu, X., & Zhang, J. (2022, February 2). Current Progress in the Extraction, Functional Properties, Interaction with Polyphenols, and Application of Legume Protein. Journal of Agricultural and Food Chemistry. American Chemical Society. https://doi.org/10.1021/acs.jafc.1c07576

Keskin, S. O., Ali, T. M., Ahmed, J., Shaikh, M., Siddiq, M., & Uebersax, M. A. (2022, March 1). Physico-chemical and functional properties of legume protein, starch, and dietary fiber—A review. Legume Science. John Wiley and Sons Inc. https://doi.org/10.1002/leg3.117

Mulla, M. Z., Subramanian, P., & Dar, B. N. (2022). Functionalization of legume proteins using high pressure processing: Effect on technofunctional properties and digestibility of legume proteins. LWT158. https://doi.org/10.1016/j.lwt.2022.113106

Drulyte, D., & Orlien, V. (2019). The effect of processing on digestion of legume proteins. Foods8(6). https://doi.org/10.3390/foods8060224

Round 2

Reviewer 1 Report

Manuscript ID: foods-2441646

Manuscript Title: Overview of the incorporation of legumes into new food options: An approach on versatility, nutritional, technological, and sensory quality

The author has made a lot of revisions to the paper, but there are still some problems.

 Comments:

1.I still think that the scope of this paper is too large, resulting in the incomplete summary of relevant research progress in each part.

2.Line 77: July 2021?

3.Line 86: (2) open access: It means that the references must be available in open access?

4.In part 2: Would a table summarizing the progress of research on the nutritional qualities of different types of legumes be better than Table 1?

5.The text of the part 3 and part 4 is suggested to be re-polished and revised, rather than the simple list of the summary of each article individually.

6. The text of future research should be revised and refined.

Author Response

Dear reviewer,

The authors greatly appreciate the new comments from the reviewers. The points mentioned by the reviewer were very helpful in one more time improving the manuscript and in delivering a better paper for readers.

We hope this 3rd revision has clarified some of the concerns raised by the reviewer.

Based on your valuable comments we tried to improve some points.

Manuscript Title: Overview of the incorporation of legumes into new food options: An approach on versatility, nutritional, technological, and sensory quality

The author has made a lot of revisions to the paper, but there are still some problems.

 Comments:

1.I still think that the scope of this paper is too large, resulting in the incomplete summary of relevant research progress in each part.

Answer:  Some readers may even think that we may have written less than they would have wished, but the authors have modified some of the content. We only approached in the paper what we worked on the last 15 years, which has been the insertion of legumes in food preparation and the impact of this use in nutritional, technological, and sensory quality in the final product.

2.Line 77: July 2021?

Answer: The authors have revised again. Thanks for your attentive eyes.

3.Line 86: (2) open access: It means that the references must be available in open access?

Answer: We have tried to improve it, according our referee suggestion.

4.In part 2: Would a table summarizing the progress of research on the nutritional qualities of different types of legumes be better than Table 1?

Answer:  Sorry, but our deadline was just three days. We apologize to our reviewer for the expectation we created. We tried to improve this part, but since this was not a systematic review, we did not intend to include all the articles that address this topic. Our objective was to illustrate and bring out the differences in the products that can be made with legumes, from hamburgers to breads, among other options. We did not include any legumes specifically because the idea was not to favor any of them, but legumes in general.

5.The text of the part 3 and part 4 is suggested to be re-polished and revised, rather than the simple list of the summary of each article individually.

Answer: We hope it is better this time. We tried to improve it.

  1. The text of future research should be revised and refined.

Answer:  You are right, it definitely needed some improvement.

Reviewer 2 Report

Dear Authors, the corrections made have greatly improved the quality of the article. Thank you for your effort.

Author Response

Dear reviewer,

The authors greatly appreciate the new comments from the reviewers. The points mentioned by the reviewer were very helpful to improve the manuscript and in delivering a better paper for readers.

We hope this new version has clarified some of the concerns raised by the reviewer.

Based on your valuable comments we tried to improve some points.

Best wishes,  VRO
